# Shared Neurodevelopmental Perturbations Can Lead to Intellectual Disability in Individuals with Distinct Rare Chromosome Duplications

**DOI:** 10.3390/genes12050632

**Published:** 2021-04-23

**Authors:** Thiago Corrêa, Cíntia B. Santos-Rebouças, Maytza Mayndra, Albert Schinzel, Mariluce Riegel

**Affiliations:** 1Department of Genetics, Institute of Biosciences, Federal University of Rio Grande do Sul UFRGS, Porto Alegre 91501-970, Brazil; thiagocorrea252@gmail.com; 2Department of Genetics, Institute of Biology Roberto Alcantara Gomes, State University of Rio de Janeiro, Rio de Janeiro 20511-010, Brazil; cbs@alternex.com.br; 3Children’s Hospital Jeser Amarante Faria, Joinville 89204-310, Brazil; maytzac@gmail.com; 4Institute of Medical Genetics, University of Zurich, 8952 Schlieren, Switzerland; schinzel@medgen.uzh.ch; 5Medical Genetics Service, Hospital de Clínicas de Porto Alegre, Porto Alegre 90035-903, Brazil

**Keywords:** duplication syndromes, intellectual disability, axon guidance, PPI-network

## Abstract

Chromosomal duplications are associated with a large group of human diseases that arise mainly from dosage imbalance of genes within the rearrangements. Phenotypes range widely but are often associated with global development delay, intellectual disability, autism spectrum disorders, and multiple congenital abnormalities. How different contiguous genes from a duplicated genomic region interact and dynamically affect the expression of each other remains unclear in most cases. Here, we report a genomic comparative delineation of genes located in duplicated chromosomal regions 8q24.13q24.3, 18p11.32p11.21, and Xq22.3q27.2 in three patients followed up at our genetics service who has the intellectual disability (ID) as a common phenotype. We integrated several genomic data levels by identification of gene content within the duplications, protein-protein interactions, and functional analysis on specific tissues. We found functional relationships among genes from three different duplicated chromosomal regions, reflecting interactions of protein-coding genes and their involvement in common cellular subnetworks. Furthermore, the sharing of common significant biological processes associated with ID has been demonstrated between proteins from the different chromosomal regions. Finally, we elaborated a shared model of pathways directly or indirectly related to the central nervous system (CNS), which could perturb cognitive function and lead to ID in the three duplication conditions.

## 1. Introduction

Chromosomal duplication syndromes are caused by intrachromosomal rearrangements (due to genomic instability) and may result in overexpression of dosage-sensitive genes within the rearrangement and gene interruption or gene fusion at the breakpoint junctions [1,2]. As a consequence, chromosomal duplications can affect multiple functional proteins that need to be effective in terms of quantity, location, and time of activity. The consequence of these alterations can lead to damage in brain development and/or cognitive functioning [3,4]. Moreover, imbalances of proteins that compose multiprotein complexes may be extremely deleterious, when stochiometric changes in subunits affect biological processes [2,5]. Finally, the perturbation of hub-genes may also alter the expression and function of other sets of proteins, or even, produce aggregation of proteins that lead to cellular toxicity [2,6].

Chromosomal duplications have a prevalence of ~0.7/10.000 births and are commonly associated with syndromic forms of Intellectual Disability (ID), a heterogeneous condition with a worldwide prevalence of 1% [4,7] that impairs intellectual functioning and adaptive behavior, manifesting before adulthood [8]. Usually, duplication syndromes are identified by Chromosomal Microarray Analysis (CMA), considered as the first-tier test that offers 15–20% of diagnostic rate for individuals with unexplained global developmental delay/ID and/or congenital anomalies [9].

Neurological processes are tightly regulated during the development and throughout the individual’s life in a manner that any change can have deleterious effects on cognitive function [10]. Many cellular processes are affected in ID, including neurogenesis, neuronal migration to axon guidance, synaptic plasticity, dendritic arborization, and regulation of transcription and translation. These bioprocesses can converge on similar and connected pathways, involving common phenotypic manifestations [4,10,11]. Pathophysiology causing ID comprises proteins that emerge in pathways and cellular networks involving several biological functions that occur through interactions represented by the human interactome [4,12]. Moreover, chromosomal rearrangements may include regions significantly enriched for genes involved in brain development that can generate multiple pathogenic mechanisms [2].

Herein we determined whether genes located in duplicated regions in three patients followed up at our genetics service with rare but relevant regions (8q24.13q24.3, 18p11.32p11.21, and Xq22.3q27.2) are involved in shared central molecular pathways associated with genes related with ID. The 8q24.13q24.3 duplication identified is a rare chromosomal rearrangement associated with dysmorphic features, growth delay, and ID [13,14,15,16]. Moreover, variable levels of ID and cerebellum hypoplasia were described in patients with 18p11 duplications, however, few cases of pure duplications in this region have been reported with similar rearrangements so far [17,18,19,20,21]. Duplication at Xq22.3q27.2 is a condition with region enriched in genes related to neurological function involving many cases of ID, behavioral problems, holoprosencephaly, and cerebellar vermis hypoplasia [22,23,24,25,26]. Therefore, we integrated several levels of data by identification of gene content, protein-protein interactions, and functional analysis on specific tissues to suggest a model with common or related pathways to the central nervous system (CNS) functions in individuals affected by these duplication syndromes.

## 2. Materials and Methods

### 2.1. Chromosomal Microarray Analysis (CMA)

Three patients with ID were followed in the Medical Genetics Service—HCPA. The duplications were mapped by CMA using a 60-mer oligonucleotide-based microarray with a theoretical resolution of 40 kb (8 × 60 K, Agilent Technologies Inc., Santa Clara, CA, USA). The labeling and hybridization were performed following the protocols provided by Agilent, 2011. The arrays were analyzed using a microarray scanner (G2600D) and the Feature Extraction software (version 9.5.1, both from Agilent Technologies). UCSC Genome Browser on Human Feb. 2009 (GRCh37/hg19) was employed to identify all protein-coding genes from duplicated regions. The complete list of protein-coding genes can be seen in Appendix A.

### 2.2. Interactome Construction and the Expanded Duplication Syndromes Interactome (eDSi)

The human interactome was generated using the Human Integrated Protein-Protein Interaction Reference (HIPPIE) database (version 2.2) [27]. We filtered in the interactions with confidence score > 0.4 and limited our analysis to the largest connected component, containing 16,108 nodes and 256,552 links/edges. Next, we extracted only protein-protein interactions from the three selected duplicated regions (Appendix A) and selected their first neighbor to expand and generate the eDSi. Cytoscape V.3.7.0. software [28] was used for visualization, and calculations of centrality parameters of the networks.

### 2.3. Functional Modules Detection and Enrichment Analysis

The HumanBase database integrates functional networks in tissues, gene expression, and disease associations. Evidence is provided by a massive set of experiments containing more than 14,000 publications and 144 tissue- and cell lineage-specific functional contexts [29,30]. We extracted significant biological processes in the eDSi, by using the detection of functional modules tool in specific tissues available in the HumanBase [31]. This tool allows the detection of tissue-specific functional modules, comprising related genes located in clusters that share local network neighborhood. The method uses k-nearest-neighbor (SKNN) and the Louvain community-finding algorithm to cluster the genes list into distinct modules of tightly connected genes [31]. q value was calculated using one-sided Fisher’s exact tests and Benjamini–Hochberg corrections to correct for multiple tests and only values < 0.05 were considered (Appendix A). Moreover, we used the webserver Enrichr [32] to identify significant pathways involved with neuronal functions in genes from the tissue-specific network. The gene-set libraries used were BioCarta, BioPlanet, Elsevier Pathway Collection, Kegg, Reactome, Panther, and WikiPathways. We considered only bioprocesses with a *p*-value < 0.05.

### 2.4. Prioritization of Candidate Genes

To prioritize candidate genes associated with ID, we used a list of known ID-genes (Appendix A), available at http://www.disgenet.org/ (accessed on 6 January 2020). DisGeNET is a platform that integrates data from UNIPROT, CGI, ClinGen, Genomics England, CTD (human subset), PsyGeNET, and Orphanet on human gene-disease associations [33]. We used a query list of ID-genes to expand the selection of nodes, using network propagation through the Diffusion algorithm (V. 1.6.1) [34]. Network propagation can estimate the distance between different sets of proteins, and identify a subnetwork with nodes closely related to each other [34]. The proximity among candidate genes and query ID-genes in the eDSi was measured using 302 as a maximum diffusion rank (highest allowed value). The complete list of prioritized genes is shown in Appendix A.

### 2.5. Functional Tissue-Specific Data

We used the list of the prioritized genes (Appendix A) to identify gene expression in tissues and construct a gene-disease association network in the HumanBase [29,30]. Moreover, a tissue-specific network with 18 genes highly expressed in the CNS was generated using data from co-expression, protein interaction, TF binding, microRNA targets, and perturbations. We prioritized the most expressed genes in the CNS, or genes previously reported in the literature involving ID in individuals with duplication regions. The parameters used to generate the network were a confidence >0.10 and a value of 15 for the maximum number of genes.

## 3. Results

### 3.1. Identification of Rare Chromosome Duplications

Chromosome duplications were mapped using the samples of three patients with ID using hg/19 reference: 8q24.13q24.3 (Chr8:126,397,316–143,577,971); dup18p11.32p11.21 (Chr18:14,316–14,773,575); and dupXq22.3q27.2 (Chrx:106,283,188–140,340,737). The summary of CMA and clinical findings from the three patients with chromosomal duplications can be seen in Appendix A and Table 1, respectively.

### 3.2. DSi Proteins Tend to Have High Values of Betweenness

The human interactome provided a network-based framework to investigate protein-protein interactions between DSi-proteins (Figure 1a). The extraction of protein-coding genes from the duplicated regions and their first neighbor resulted in a DSi composed of 3016 nodes/proteins and 4330 links/interactions (Figure 1b). DSi included 89 proteins from duplicated regions and 65 ID-genes. Four DSi-proteins (LAMA1, STAG2, NKAP, and ALG13) were also found among the ID-genes list [35].

The average centrality measures in the human interactome were: degree (31.85), betweenness (0.00013), closeness (0.3249), and shortest path length (3.119). Degree centrality defines the number of connections of a specific node in the network, and in the biological context, nodes with a degree value > 100 links (hubs) may have multiple functions in cellular networks [36]. Betweenness corresponds to the number of nonredundant shortest paths that pass through a node of interest and may indicate the potential of a protein to create a bridge for communication between distant nodes [37,38]. The average shortest path length involves the summa of all shortest paths between nodes couples, divided by all pairs of nodes in the network, and the closeness indicates how close a node is to all other nodes in the network [39,40].

Many DSi-proteins showed higher values of centrality, compared to the mean of the human interactome, indicating topological relevance to specific DSi-proteins (Figure 1c). In this sense, 14% of DSi-proteins were considered hubs, including MYC, a transcription factor, and CUL4B, a central component of the ubiquitin-protein ligase complex, both acting in several biological processes. Moreover, other hubs, such as RBMX, PTK2, AIFM1, VAPA, and XIAP, are associated with ID [41,42,43,44,45].

Eighty percent of the DSi-proteins reached a betweenness centrality value higher than the average of the human interactome (Figure 1c). VGLL1 (coactivator for the mammalian TEFs), CDR1 (neuronal signal transduction protein), MC5R (melanocortin receptor coupled to the transmembrane G protein), and WISP1 (a member of the WNT1 inducible signaling pathway) showed high betweenness values. All of these proteins have significant roles in signal transduction or coactivation of transcription factors [46,47,48,49]. Furthermore, CDR1 is a putative neuronal protein identified in individuals with cerebellar degeneration [50].

Besides degree and betweenness, shortest path length and closeness were calculated. VGLL1, CDR1, WISP1, and MC5R also emerged in the network with high closeness and lower shortest path length values (Figure 1c). From a biological perspective, these nodes can have a major impact on proteins that are close to the node or serve as the shortest path among distant proteins in the network. About 22% of the proteins were identified with values of closeness and shortest path length above the average of the interactome. The main results of the topological characteristics of other DSi-proteins can be seen in Figure 1c,d.

### 3.3. Biological Processes Associated with Rare Duplications

We carried out enrichment analysis of the DSi-proteins to identify biological processes with a possible role in ID. Six clusters were detected grouping the main bioprocesses (Figure 2). Cluster 1 identified only proteins from dup18p11.32p11.21 with enriched bioprocesses related to chromosome segregation. DSi-proteins from 8q24.13q24.3, 18p11.32p11.21, and Xq22.3q27.2 were found in clusters 2 and 6, associated with telomere maintenance, DNA repair, epithelium developmental, and ion transport. Cell morphogenesis in clusters 3 and 4 was associated with proteins from duplicated regions on chromosomes 8, 18, and X. Cluster 5 is the only one to encompass proteins from the three duplicated regions, with enrichment for microtubule cytoskeleton organization, negative regulation of cell cycle, and neurogenesis. Cell pathways involving the ID pathophysiology can encompass changes in the cytoskeleton dynamics, neurogenesis, and morphology during synaptic plasticity or neuronal development [11,51].

In addition, plasma membrane-bounded cell projection was enriched in the three duplicated regions. This process involves the formation of a prolongation bounded by the plasma membrane, such as an axon. Projection defects were reported in an ID mouse model [52]. Moreover, neuronal development and nuclear chromosome segregation were identified in functional enrichment analysis of ID-genes and DSi-genes from dupXq22.3q27.2 and dup18p11.32p11.21. The complete results are provided in Appendix A.

### 3.4. DSi-Genes Are Widely Expressed in the CNS

ID is caused by perturbations in the significant biological functions that impact cellular networks present in different regions of the CNS. We identified the influence of each of the 44 prioritized genes (Appendix A) in different tissues and found that these genes are mainly expressed in the CNS when compared with other tissues in humans (Figure 3) [14,16,53].

Therefore, we extracted expression data from multiple CNS regions to better understand the influence of each gene on this tissue (Figure 4). Many genes located at Xq22.3q27.2 are widely expressed in the CNS and were previously associated with syndromic/non-syndromic X-linked ID, such as ALG13, PAK3, THOC2, GRIA3, STAG2, OCRL1, AIFM1, PHF6, RMBX, SOX3, LAMP2, CUL4B, and UBE2A [54,55]. Moreover, patients with duplicated regions that encompass the X-linked genes SOX3, STAG2, AIFM1, GRIA3, PAK3, and OCRL exhibit ID [22,23,24,25,26,56,57,58]. Moreover, six genes from the duplicated region 18p11.32-p11.21 are highly expressed in several regions of the CNS, from which three of them (LAMA1, MYOM1, and TGIF1) were duplicated in individuals with ID [18,19,20]. Furthermore, patients with duplication of 8q24.13q24.3 region involving the KCNQ3, PTK2, ASAP1, and NDRG1 genes, which are widely expressed in CNS, presented ID [14,16,53].

### 3.5. Candidate Proteins from Different Chromosome Rearrangements Interact with Each Other in the CNS Network

To analyze the relevance of candidate proteins according to tissue specificity, we constructed a network with interactions from the CNS, in an attempt to identify clues about the likely contribution of each protein in the development of ID. The CNS network includes 32 nodes connected by 210 interactions, from which 18 are DSi proteins (Figure 5a). The most connected proteins are PTK2 (19), STAG2 (16), and TGIF1 (16). Interestingly, ID-genes WAC, QKI, and PPP1R12A emerge as interacting factors on the network by automatic addition of the database. It is worth mentioning that many links in the tissue-specific network result from gene co-expression studies in the context of neurological conditions, such as recessive X-linked dystonia-parkinsonism, Rett syndrome, and Huntington’s disease.

The functional enrichment analysis identified several pathways associated with axon guidance (Figure 5b). The genes directly involved with this biological process include PTK2 and KCNK3 (dup 8q24.13q24.3), LAMA1 (dup 18p11.32p11.21), and PAK3, DCX, SOX3, and OCRL (dupXq22.3-q27). As mentioned above, all these genes have already been identified in duplicated regions in individuals with ID. Moreover, LAMA1 was also present in our ID-list which used the candidate genes prioritization. Pathways related to functions necessary to axon guidance that encompasses these genes, include actin cytoskeleton regulation (*p* = 0.0060), L1CAM interactions (*p* < 0.0001), EPH-ephrin signaling (*p* = 0.0107), signaling by Rho GTPases (*p* = 0.0029) and MET cell motility promotion (*p* = 0.0003). However, other fundamental pathways in the axon guidance context can be seen in Figure 5b. The PPP1R12A gene, added to the database, is the only one not belonging to the duplicated region that appears in the ID-genes list and is involved in axon guidance.

Taken together, these results indicate that genes from different duplicated regions may be related to each other and other genes previously associated with ID localized in cellular networks in the nervous tissue and involved in neurodevelopment processes (Figure 5c).

### 3.6. Candidate Genes Are Associated with the ID

Similar pathways are disrupted in ID and in other neurological diseases due to the functional relationships of genes located in the same module in the human interactome. Therefore, to identify DSi-genes implicated in other neurological diseases, and help to confirm our results, we generated a gene–disease association network (Figure 6).

The most common diseases or phenotypes found on the network were autism spectrum disorder, peripheral CNS disease, and ID with 43, 17, and 15 associated genes, respectively. Maximum scores between disease and genes were seen in brain disease, holoprosencephaly, syndromic/non-syndromic ID, and syndromic/non-syndromic X-linked ID. The genes with the highest number of connections with other diseases were PAK3, GRIA3, and ADGRB1 associated with eight, six, and six neurological diseases, respectively. As expected, these genes were the most expressed in the CNS tissue (Figure 3). Previous data support the known relationships of many genes on the network with neurological diseases, especially located in Xq22.3-q27.2 [55]. Moreover, the candidate genes PAK3, OCRL, DCX, PTK2, KCNQ3, SOX3, and LAMA1 were associated with autism, brain disease, Dent disease, and other conditions that present ID as a hallmark, corroborating our findings (Figure 6).

## 4. Discussion

Genomic disorders caused by duplications of chromosome segments confer potential risk of global developmental delay and ID, impacting the IQ, and educational achievement of individuals [59,60,61]. The imbalance in gene dosage caused by chromosomal duplications can destabilize several genes by spreading through interactions in cellular subnetworks during neurodevelopment. Moreover, the chromosome rearrangements identified in our patients are rare, with few cases reported so far. These duplicated regions have been reported as pathogenic and ID is a recurrent clinical finding in the affected individuals [16,21,26]. Therefore, we used network analysis in an attempt to identify the potential sharing of biological processes and genes responsible for the pathophysiology of ID in rare duplications. We found seven candidate genes: *PTK2* and *KCNK3* (dup 8q24.13q24.3), *LAMA1* (dup 18p11.32p11.21), and *PAK3*, *DCX*, *SOX3*, and *OCRL* from dupXq22.3q27, all duplicated in individuals with ID [15,16,19,23,25,55,56,57,61]. Furthermore, all candidate genes identified have been reported in duplicated regions of several ID patients in the web-based database—DECIPHER.

*PTK2*, protein tyrosine kinase 2, emerged with high degree and betweenness values (hub-bottleneck) through topological analysis in the eDSi (Figure 1c). This result correctly reflects the many biological functions performed by *PTK2* that involve the regulation of migration, adhesion, protrusion, and proliferation of the cell. Besides that, *PTK2* promotes axon growth and guidance and synapse formation during CNS development [62,63,64,65]. Therefore, changes in *PTK2* expression can impair brain development and lead to mental conditions [66]. Our topological analysis supports the identification of candidate disease genes that tend to be more central to the network, and not in peripheral regions as we expected [67,68]. Moreover, we identified many proteins from duplicated regions with high betweenness values considered bottlenecks, essential nodes in the information flow between distant proteins in cellular networks [38], indicating a potential impact in pathophysiology, when dysregulated.

We identified significant expression of duplicated genes in the CNS conversely to other tissues (Figure 3). Moreover, candidate genes present remarkable expression in regions of CNS associated with ID (Figure 4), such as the cortical region and the cerebellum [10]. Candidate genes from different chromosomes interact with each other in the tissue-specific network, demonstrating functional relationships among these genes in the CNS. For instance, *PTK2* (chr:8) interacts directly with *OCRL* (inositol polyphosphate-5-phosphatase—chr:X), *DCX* (doublecortin—chr:X) with *LAMA1* (laminin subunit α 1—chr:18), or yet, *KCNQ3* (potassium voltage-gated channel subfamily Q member 3—chr:8) and *LAMA1* are connected to each other by the neurotrophic tyrosine receptor kinase (*NTRK3*) (Figure 5a). Furthermore, *OCRL* and *PTK2* interacts directly with *WAC*, *QKI*, and *PPP1R12A*, genes previously associated with ID [69,70,71,72]. In the case of *PPP1R12A*, its loss-of-function causes holoprosencephaly and ID in individuals with stop gain variants and deletions/duplications, resulting in a frameshift effect [72]. PPP1R12A protein is present in pathways, such as RHO actin cytoskeleton regulation, ROCKs activation by GTPases, dendritic spine morphogenesis, and stabilization, all bioprocesses directly or indirectly involved with axon guidance mechanisms.

Axon guidance was the most enriched term in the tissue-specific network, besides the identification of various signaling pathways directly or indirectly involved in this biological process (Figure 5b). The axon guidance process plays an essential function in neuronal wiring in the developing spinal cord, where it is responsible for extending axons and reaching their targets to form synaptic junctions. These mechanisms allow the connection between the central and peripheral nervous system during neurodevelopment, through extracellular and transmembrane molecules and their cell surface receptors [73,74,75]. The main axon guidance pathways and mechanisms involving our candidate genes were schematized in Figure 5c. The disruption or disintegration of neural circuit formation during CNS development affects cognitive function and can result in mental conditions such as ID [76,77,78]. The current model of the axon orientation mechanism reveals that the expression of guidance receptors occurs in the growth cone to indicate their targets and allow migration by controlling attractive and repulsive forces containing many guidance molecules present in their environment [75,77]. Therefore, the model of neural circuit formation supports the idea that changes in gene dosage caused by chromosomal duplications may impair the balance of this mechanism during the CNS development [79], where the gain or loss-of-function can impair the tight regulation of gene sets and cause disturbances in neighbor proteins in networks. However, expression data from patients with these chromosomal duplications should be used to confirm this model.

We observed interactions in the gene-disease association network between neurological conditions with ID and DSi-genes of three different chromosomes (Figure 6). These data suggest that duplicated regions could generate perturbations and propagate through modules in the interactome associated with many diseases linked to the CNS. For instance, the partial duplication of the gene that encodes the neuronal development transcription factor SOX3 can cause impairment in pituitary development and cognitive functions [80]. *PAK3* is expressed in the brain, playing a role in the control of cytoskeleton regulation, cell migration, axonal guidance, and synaptic plasticity, while its deregulation causes neurological abnormalities, such as ID [81,82]. *PAK3* pathogenic variants in affected males were associated with spatial cognitive abilities, defects in attention, and speech difficulties [83,84], and a hemizygous missense variant in this gene was found in two male siblings with ID [85]. *OCRL* regulates the traffic in the endosomal machinery and its depletion affects the recycling of various classes of receptors [86]. Dent disease patients with pathogenic variants in the *OCRL* can present mental impairment [87,88]. Already, *DCX* plays a crucial role in the CNS, enhancing the axonal outgrowth in postnatal cortical neurons [89]. Variants in *DCX* result in X-linked lissencephaly in males, and its overexpression leads to destabilization of microtubules and inhibition of neurite outgrowth [90]. Beyond the *PTK2* gene (a duplicated region on chromosome 8), *KCNQ3* encodes a protein with functions in the regulation of neuronal excitability and plasticity [91,92]. Pathogenic variants in this gene were identified in patients with early-onset epilepsy and neurocognitive deficits [93]. Moreover, a gain of function variants in *KCNQ3* causes neurodevelopmental delay and autistic features [94]. Lastly, *LAMA1* (duplicated region on chromosome 18), laminin involved in cell adhesion and axon outgrowth during embryonic development is associated with cerebellar dysplasia and ID in individuals with homozygous variants [95,96,97,98].

The phenotype in these conditions is not only the result of deficient protein, but also perturbations that spread in the cellular networks. Therefore, the network-based analysis, regardless of the origin of the pathogenesis of chromosomal duplications (epigenetic alteration, gain of function, effect of position, change of transcription factor sites, or deregulation of miRNAs), can help to predict the consequence of these mechanisms by analyzing functional protein relationships and their interactions in a network [99,100].

## 5. Conclusions

We found functional relationships among genes from three different duplicated chromosomal regions, reflecting interactions of protein-coding genes and their involvement in common cellular subnetworks. Furthermore, the sharing of common significant biological processes associated with ID has been demonstrated between proteins from the different chromosomal regions. According to our results, we indicate potential molecules and signaling pathways responsible for neuronal wiring that can be deregulated during neurodevelopment and cause ID. Further analysis of gene expression would be necessary to generate experimental data for these conditions in order to show more evidence regarding the association between gene expression and ID.

## Figures and Tables

**Figure 1 genes-12-00632-f001:**
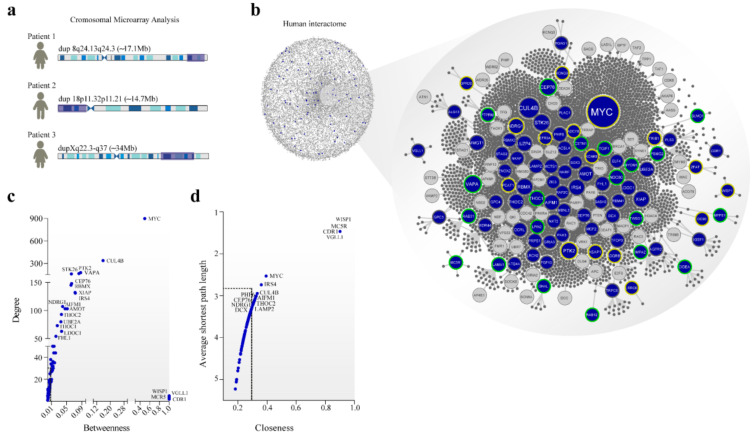
Cytogenetic to the interactome. (**a**) left: duplicated regions mapped by CMA in the patients; right: curated human interactome composed of 13,460 proteins and 141,296 interactions. In the network, proteins are nodes connected by interactions. (**b**) expanded duplication syndromes interactome (eDSi) with 3016 proteins and 4330 interactions. Blue nodes are protein-coding genes from duplicate regions; grey nodes are ID-genes. Node border colors represent the origin of duplication: dup 8q24.13q24.3 in yellow; dup 18p11.32p11.21 in green; dupXq22.3-q27 in grey. Node size is related to the number of connections (degree). (**c**) Topological parameters with degree and betweenness distribution and values of (**d**) average shortest path length and closeness for duplicated protein-coding genes in eDSi. Dashed lines in black indicate the average of these parameters for human interactome.

**Figure 2 genes-12-00632-f002:**
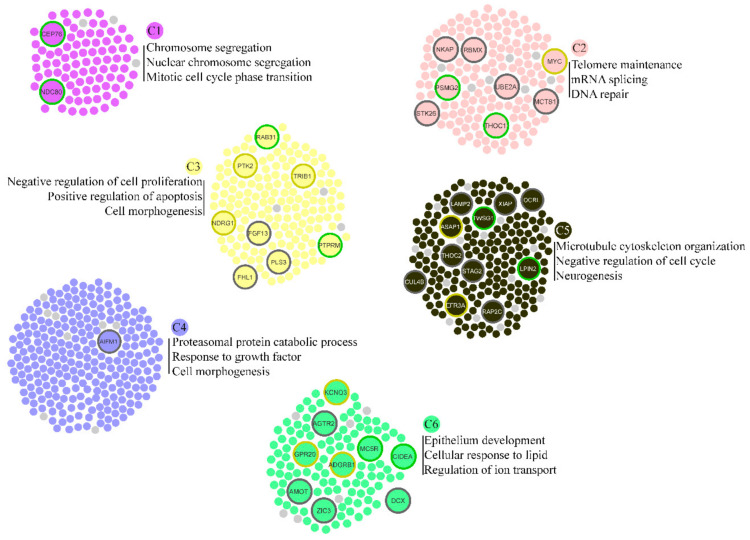
Functional cluster detection in eDSi. Node border colors represent the origin of duplication: dup 8q24.13q24.3 in yellow; dup 18p11.32p11.21 in green; dupXq22.3-q27 in grey. Small nodes in light grey represent ID-genes.

**Figure 3 genes-12-00632-f003:**
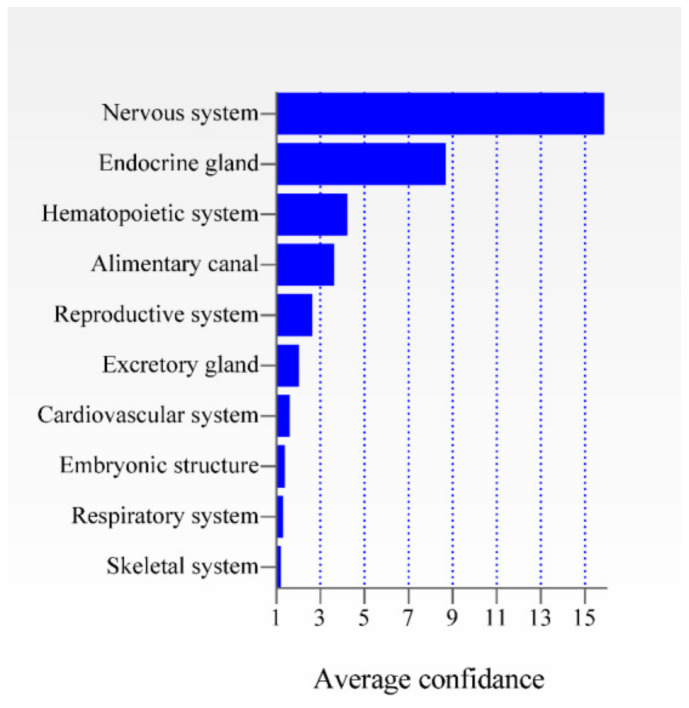
Expression of the 44 prioritized genes in ten different tissues. The average confidence value is shown for each tissue.

**Figure 4 genes-12-00632-f004:**
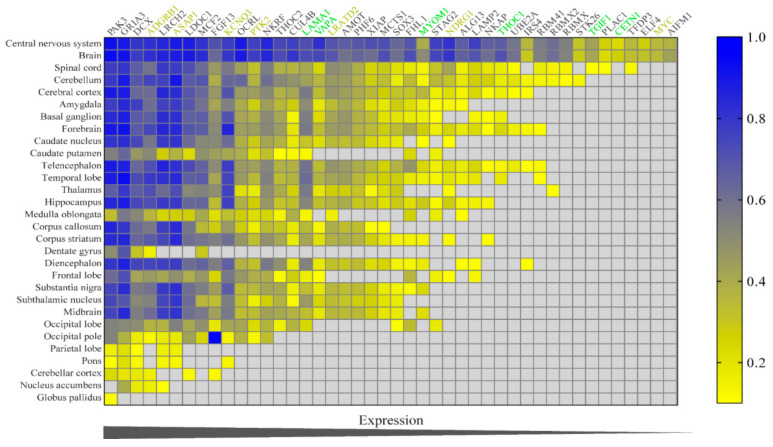
Heat map of expression of the 44 prioritized genes in CNS. Confidence value is calculated between 0–1.

**Figure 5 genes-12-00632-f005:**
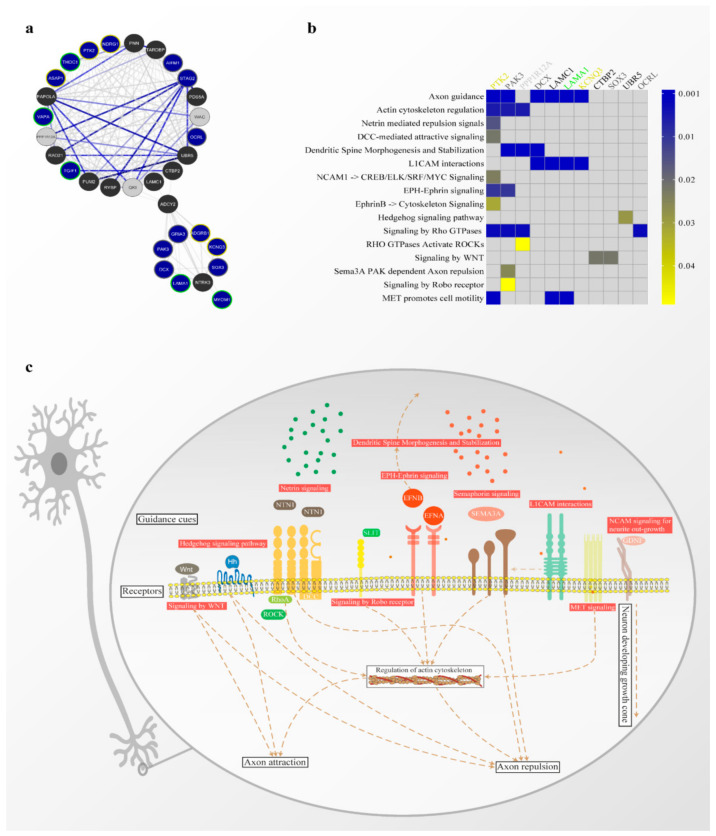
Tissue-specific gene network analysis. (**a**) functional network from CNS. Blue nodes are protein-coding genes from duplicate regions; grey nodes are ID-genes; black nodes were added by the database. Node border colors represent the origin of duplication: dup 8q24.13q24.3 in yellow; dup 18p11.32p11.21 in green; dupXq22.3-q27 in grey. Score values are shown proportionally by the thickness and intensity color of the edges. (**b**) heat map of genes from tissue-specific gene network with the most significant biological processes. (**c**) Scheme depicting the main pathways and molecules involved in axon guidance.

**Figure 6 genes-12-00632-f006:**
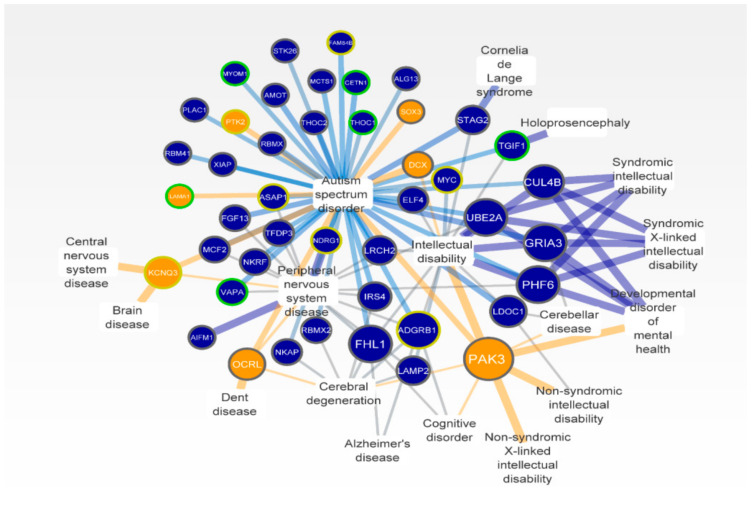
Gene–disease associations network. Duplicated genes (circular nodes) and disease of CNS (rectangular nodes). The size of the circle nodes corresponds to the number of diseases associated. Blue nodes are protein-coding genes from duplicate regions; orange nodes are candidate genes. Node border colors represent the origin of duplication: dup 8q24.13q24.3 in yellow; dup 18p11.32p11.21 in green; dupXq22.3-q27 in grey. Scores values are shown proportionally by the thickness and intensity color of the edges. Orange edges show interactions of candidate genes.

**Table 1 genes-12-00632-t001:** Summary of the CMA and clinical findings from the 3 patient with chromosomal duplications.

Patient	1	2	3
Sex	F	M	F
Age (years) *	8	12	7
Band location (duplicated)	Chr8 (q24.21-q24.3)	Chr18 (p11.32-p11.21)	ChrX (q22.3-q27.1)
CMA Deletion size (pb)	17,180,656	14,759,260	34,057,550
Genomic position (GRCh38/hg38)	Chr8:126,397,316–143,577,971	Chr18:14,316–14,773,575	Chrx:106,283,188–140,340,737
Clinical findings	ID; microcephaly; seizures; speech delay; global developmental delay	ID; speech delay; anxiety; learning difficulty; psychomotor agitation	ID; NPMDD; short stature; clinodactyly; blepharophimosis

ID: Intellectual disability; NPMDD: neuropsychomotor development delay. * age in years at the time of the CMA investigation.

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
