# Peer review of "Shared Neurodevelopmental Perturbations Can Lead to Intellectual Disability in Individuals with Distinct Rare Chromosome Duplications"

_genes, 2021, doi:10.3390/genes12050632_

Round 1
Reviewer 1 Report
Corrêa et al provided an analysis of interactions of protein-coding genes included in duplicated genomic regions identified in patients with intellectual disability. They defined functional relationships of these genes and identified candidate genes for the patients’ phenotype.
Although using an original approach, this work does not help to understand the pathophysiology of the duplications. Indeed, the duplicated regions are very large and contain a lot of gene. The results regarding intellectual disability genes and central nervous system genes may have been obtained by chance. These results should be compared to non-pathogenic duplication and to genomic simulations. The identified candidate genes belong to the duplicated regions and were for most of them already involved in intellectual disability. No common genes for the phenotype was really identified. Moreover, many candidate genes were involved in haplo-insufficiency or recessive diseases. There is no argument that a gain of copy could be pathogenic for these genes. The author did not take into account dosage sensitivity or translational regulation mechanisms.
Minor comments :
-Were the genomic gains identified by chromosomal microarray further characterized by conventional methods (karyotype/FISH), to distinguish between duplication or insertion mechanism. This could have an impact on gene expression and the pathophysiology.
-The figure 2 is not readable do to undistinguishable colours
Author Response
Reply to Reviewer #1: Thanks for your comments and suggestions.
Comment # 1 Although using an original approach, this work does not help to understand the pathophysiology of the duplications. Indeed, the duplicated regions are very large and contain a lot of gene. The results regarding intellectual disability genes and central nervous system genes may have been obtained by chance.
Response to Reviewer’s comment # 1: Please consider that we did not claim that our work would lead/ “….to understand the pathophysiology of the duplications.” This was not our aim in this specific proposal. Any other single work could and would lead to fully understand it and could b publish somewhere else. We are exploring the content of coding genes involved in the duplications and a possible shared interaction between them (relationships among genes from these three different duplicated chromosomal regions).
“…regarding intellectual disability genes and central nervous system genes may have been obtained by chance…”, consider that we use strict cut-off points to generate the data. The networks were generated considering interactions with a minimum score of 0.4 (evidence from cured experimental data). Related interactions in the context of the nervous system were scored with the highest score by the specific tissue bank. The scoring genes associated with ID in the specific tissue network (gray nodes) are not duplicated. In addition, genes involved in pathways related to the functioning of the nervous system were enriched (adj p <0.05) in both networks (see Figures 2b and 5a), showing statistical significance compared to the reference genome.
Comment # 2 These results should be compared to non-pathogenic duplication and to genomic simulations
Response to Reviewer’s comment # 2: Thank you, we added the information regarding your comment in the discussion section.
Comment # 3 The identified candidate genes belong to the duplicated regions and were for most of them already involved in intellectual disability. No common genes for the phenotype was really identified.
Response to Reviewer’s comment # 3: Yes, we partially agree with your comment. It is true that some of the duplicated genes have already been associated with ID. However, I believe that this should not be an argument for invalidate our study. Please consider that we did not claim for the identification of common genes. We described candidate genes that have not been previously associated with ID in the context of different duplications. The genes involved in the pathways, are affected by different chromosome imbalances/originated from different homologues. Thus, we find common interactions in "cellular subnetworks" related to the phenotype.
Comment #4: Moreover, many candidate genes were involved in haplo-insufficiency or recessive diseases. There is no argument that a gain of copy could be pathogenic for these genes. The author did not take into account dosage sensitivity or translational regulation mechanisms.
Response to Reviewer’s comment # 4: Yes, we agree with your comment. We struggled to find models with duplications of candidate genes, but several ID patients with duplicated region encompassing our candidate genes are registered with DECIPHER In addition, the topological importance of the protein in the network and the involvement in certain processes biological factors that require fine regulation such as "axon guidance" indicate that the slightest imbalance in the expression of these genes during development may affect the correct functioning of this pathway. We considered your comment and information were added to the article's discussion.
Comment #5: Were the genomic gains identified by chromosomal microarray further characterized by conventional methods (karyotype/FISH), to distinguish between duplication or insertion mechanism. This could have an impact on gene expression and the pathophysiology
Response to Reviewer’s comment # 5: Yes. We confirmed that the genomic gains were investigated by cytogenetic conventional methods and they are duplications. The cytogenetic characterization of the rearrangements are not the focus of the present study. We are exploring the content of coding genes involved in the duplications and a possible shared interaction between them (relationships among genes from these three different duplicated chromosomal regions). Thus, due to the space limitation we are showing the most important and relevant date that supports the main focus of the present study. However, the summary of the CMA can be seen in Table 1 and S1.
Comment #6: The figure 2 is not readable do to undistinguishable colours
Figure 2 was modified for better visualization. Please also consider that in case the manuscript is accepted for pubication, the original archives can be adapted by the graphic editing team according to the edition style (Space, size and figures configuration) of the journal.
Thank you in advance for your consideration to our revised version of the manuscript. Please consider that this is an original network-based study, an approach which is still uncommon to explore chromosomal imbalance rearrangements regions associated with ID. The work is a documented evaluation of the efficiency of network-based analysis to study ID associated with chromosomal duplications. We have been demonstrating in recent publications(https://doi.org/10.1002/ajmg.c.31900;https://doi.org/10.3389/fgene.2020.00561; http://orcid.org/0000-0002-6397-1301), that the contribution of the network-based analysis is that, regardless of the origin of the pathogenesis of a deletion/duplication syndrome (epigenetic alteration, loss of function of the deleted proteins, effect of position, change of transcription factor sites or deregulation of miRNAs), it considers all these mechanisms when estimating how the proteins interact or fail to interact with each other in the interactome. Therefore, we propose that our manuscript may contribute to understand the impact of genomic imbalances sharing common significant biological processes associated with ID.
Further, we believe that this manuscript will be of interest to the readership of the journal because systems biology approaches allow us to broaden our understading of contiguous gene imbalances, and the approach reported here provides one more tool for the investigation of ID.
Reviewer 2 Report
The work is very interesting, well structured and nicely written. The authors proposed a genomic comparative delineation of genes located in duplicated chromosomal regions: 8q24.13q24.3, 18p11.32p11.21, and Xq22.3-q27.2 in three patients, which has the ID phenotype as a common feature. They studied the gene content within the duplications, protein-protein interactions, and functional analysis on specific tissues.
They discovered functional relationships among genes from these three different duplicated chromosomal regions, they underlined the interactions of protein-coding genes and their involvement in common cellular subnetworks. Moreover, the sharing of common significant biological processes connected with ID has been shown between proteins from the aforementioned chromosomal regions. The authors also developed a shared model of pathways associated with the central nervous system, which could cause ID in the three conditions . Furthermore, they indicate potential molecules and signaling pathways responsible for neuronal wiring that can be deregulated during neurodevelopment and cause ID.
This work is very complete and will be useful in order to understand the phenotype associated with all duplications. It would be helpful for studying the numerous cases that clinicians meet in clinical genetics, even though it will be very difficult to apply in practical genetics owing to its very complexity. However, this study is clearly fascinating and, hopefully, in the future will have many practical applications.
Concerning the study design:
- Would it be possible for the authors to provide further refinement/validation of their results by using a different set of tools, at least in some of the analytical phases? E.g. a parallel evaluation of protein-protein interactions using STRING, usage of different databases for gene expression… If the authors consider these steps unnecessary or inappropriate, could they provide a reason why?
- For better clarity, the concepts of degree, betweenness, closeness and shortest path length might warrant a short explanation as soon as they are introduced (lines 158-159) rather than in later paragraphs. A similar approach may be considered for “bottlenecks”: the explanation given at lines 322-325 is remarkably clear and well-contextualized, while its first mention at line 170 is confusing.
- Fig.1, 2 and 5 are too small to read (lettering, node borders).
- All gene names throughout the manuscript should be written in Italics.
We also recommend that the following minor modifications be carried out:
line 190: “We carried out” rather than “carry”
lines 194 and 216: “found” rather than “founded”
line 204 It would be preferable if the authors wrote "Projection defects..." instead of "Projections defects..."
line 237: “chromosome rearrangements” rather than “rearranges”
line 255 It would be preferable if the authors wrote "Score values..." instead of "Scores values..."
line 303: “risk of” rather than “to”
line 311: “candidate genes”: missing some punctuation after “genes”
line 331: “interacts” rather than “interact”
line 343: “directly” rather than “direct”
lines 345-346: “for extending axons and reaching” rather than “by extend axons and reach”
line 351: “conditions such as” rather than just “as”
line 358 The authors should write "....tight regulation.." instead of "...tightly regulation.."
line 359 It would be preferable if the authors wrote "...neighbor proteins..” instead of “…neighbours proteins…”
N.B. The authors use American spelling such as in "color" so it is respectfully suggested that, in order to be consistent, they adopt American spelling in "neighbor" as opposed to the British "neighbour".
line 369: “attention” rather than “attentional”
and the usage of articles throughout the text.
On the other hand, some sentences should be rewritten to clarify their meaning. In particular, I would like to point out the following ones (the most problematic passages were underlined):
lines 21-23 “Here, we report... which has... common hallmark”
lines 37-39: “proteins that.. causing among distinct effects… functioning”
lines 54-62: “Many cellular processes… underlie common phenotypic… through interactions represented by the human interactome… included regions… pathogenic mechanisms.”
lines 65-66: “molecular pathways, to which… (verb?)… with ID”
line 71: “Duplications at… is a region enriched in genes…”
lines 275-276: “Similar pathways… due to genes functionally relationships”
lines 333-334: “or yet, … connected through of neurotrophic…”
lines 352-355: The current… growth cone by the axons… migrate by control… environment”
After the above modifications have been effected, we believe that the article should be accepted for publication.
Author Response
Reply to Reviewer #2: Thanks for your comments and suggestions.
Comment #1:Would it be possible for the authors to provide further refinement/validation of their results by using a different set of tools, at least in some of the analytical phases? E.g. a parallel evaluation of protein-protein interactions using STRING, usage of different databases for gene expression… If the authors consider these steps unnecessary or inappropriate, could they provide a reason why?
Response to Reviewer’s Comment # 1: Human Integrated Protein-Protein Interaction Reference (HIPPIE) integra PPIs de múltiplos datasets experimentais e curados. In HIPPIE “Several resources have been created that, like HIPPIE, integrate PPI data from multiple sources but do not have a focus on distributing a simple scored dataset, while offering excellent tools to examine evidence behind each PPI (e.g. iRefWeb [34]) or do not focus on experimentally verified interactions (e.g. STRING [35])” (Alanis-Lobato et al, 2017). “The HumanBase database integrates functional networks in tissues, gene expression, and disease associations. Evidence is provided by a massive set of experiments containing more than 14,000 publications and 144 tissue- and cell lineage-specific functional contexts” (linhas 100-102). Furthermore, collection and tissue methods used may change between databases. Some of them using samples of cadavers, with different period between death and the collection of biological material.
Comment #2: For better clarity, the concepts of degree, betweenness, closeness and shortest path length might warrant a short explanation as soon as they are introduced (lines 158-159) rather than in later paragraphs. A similar approach may be considered for “bottlenecks”: the explanation given at lines 322-325 is remarkably clear and well-contextualized, while its first mention at line 170 is confusing.
Response to Reviewer’s Comment # 2: Revised as requested.
Comment #3: Fig.1, 2 and 5 are too small to read (lettering, node borders).
Response to Reviewer’s Comment # 3: Suggestion accepted. Figures were modified for better visualization. Please also consider that in case the manuscript is accepted, the original archives can be adapted by the graphic editing team according to the edition style (Space, size and figures configuration) of the journal.
Comment #4: All gene names throughout the manuscript should be written in Italics.
revised as requested
Comment #5: We also recommend that the following minor modifications be carried out:
line 190: “We carried out” rather than “carry”
revised as requested
Comment #6: lines 194 and 216: “found” rather than “founded”
revised as requested
Comment #7: line 204 It would be preferable if the authors wrote "Projection defects..." instead of "Projections defects..."
revised as requested
Comment #8: line 237: “chromosome rearrangements” rather than “rearranges”
revised as requested
Comment #9: line 255 It would be preferable if the authors wrote "Score values..." instead of "Scores values..."
revised as requested
Comment #10: line 303: “risk of” rather than “to”
revised as requested
Comment #11: line 311: “candidate genes”: missing some punctuation after “genes”
revised as requested
Comment #12: line 331: “interacts” rather than “interact”
revised as requested
Comment #13: line 343: “directly” rather than “direct”
revised as requested
Comment #14: lines 345-346: “for extending axons and reaching” rather than “by extend axons and reach”
revised as requested
Comment #15: line 351: “conditions such as” rather than just “as”
revised as requested
.Comment #16: line 358 The authors should write "....tight regulation.." instead of "...tightly regulation.."
revised as requested.
Comment #17: line 359 It would be preferable if the authors wrote "...neighbor proteins..” instead of “…neighbours proteins…”
Revised as requested.
Comment #18: N.B. The authors use American spelling such as in "color" so it is respectfully suggested that, in order to be consistent, they adopt American spelling in "neighbor" as opposed to the British "neighbour".
Suggestion accepted.
Comment #19: line 369: “attention” rather than “attentional”
Revised as requested.
Comment #20: …..and the usage of articles throughout the text. On the other hand, some sentences should be rewritten to clarify their meaning. In particular, I would like to point out the following ones (the most problematic passages were underlined):
-lines 21-23 “Here, we report... which has... common hallmark”
Sentence revised.
-lines 37-39: “proteins that.. causing among distinct effects… functioning”
Revised as requested.
-lines 54-62: “Many cellular processes… underlie common phenotypic… through interactions represented by the human interactome… included regions… pathogenic mechanisms.”
Revised as requested.
-lines 65-66: “molecular pathways, to which… (verb?)… with ID”
Revised as requested.
-line 71: “Duplications at… is a region enriched in genes…”
Revised as requested.
-lines 275-276: “Similar pathways… due to genes functionally relationships”
Revised as requested.
-lines 333-334: “or yet, … connected through of neurotrophic…”
Revised as requested.
-lines 352-355: The current… growth cone by the axons… migrate by control… environment”
Revised as requested.
Comment #21: After the above modifications have been effected, we believe that the article should be accepted for publication.
Response to Reviewer’s # 21 comment: Thank you in advance for your consideration to our revised version of the manuscript.
Reviewer 3 Report
In this manuscript Correa T et al demonstrated that genomic disorders caused by duplications in chromosomes confer potential risk to global development delay intellectual disability and multiple congenital abnormalities.
Authors provided genomic comparative delineation of genes located in duplicated chromosomal regions (8q24.13q24.3, 18p11.32p11.21 and Xq22.3-q27.2 and found seven candidate genes PTK2 and KCNK3 (dup 8q24.13q24.3), LAMA1 (dup 18p11.32p11.21), 311 and PAK3, DCX, SOX3, and OCRL from dupXq22.3-q27.
However, other concerns should be addressed to strengthen the overall quality of the paper.
- The authors should describe more clinical information about the three patients analyzed in this article
- The authors did not show the 8q24.13q24.3, 18p11.32p11.21 and Xq22.3-q27.2 results in G-band, FISH, CMA, SNP or qPCR.
- In figure 1, it’s complicate for the interpret and segregate the authors’ findings with the database. Show separate CMA results for the three patients with duplicate genomic region.
- Page 2, line 49 and line 78: Chromosomal Microarray Analysis (CMA)
without basic information, it is not possible to draw a conclusion about genomic delineation of genes.
Author Response
Reply to Reviewer #3: Thanks for your comments and suggestions.
Comment #1: The authors should describe more clinical information about the three patients analyzed in this article
Response to Reviewer’s comment # 1: Suggestion accepted. A table was added in the manuscript.
Comment #2: The authors did not show the 8q24.13q24.3, 18p11.32p11.21 and Xq22.3-q27.2 results in G-band, FISH, CMA, SNP or qPCR.
Response to Reviewer’s comment # 2: We agree with your comment. We confirmed that the genomic gains were investigated by cytogenetic conventional methods such G-band, FISH and CMA and, that they are duplications. Please consider that the cytogenetic characterization of the rearrangements are not the focus of the present study. We are exploring the content of coding genes involved in the duplications and a possible shared interaction between them (relationships among genes from these three different duplicated chromosomal regions). However, the summary of the CMA can be seen in Table 1 and S1.
Comment #3: In figure 1, it’s complicate for the interpret and segregate the authors’ findings with the database. Show separate CMA results for the three patients with duplicate genomic region.
Response to Reviewer’s comment # 3: Suggestion accepted. Figure 1 was modified for better visualization. Please also consider that in case the manuscript is accepted, the original archives can be adapted by the graphic editing team according to the edition style (Space, size and figures configuration) of the journal.
Comment #4: Page 2, line 49 and line 78: Chromosomal Microarray Analysis (CMA)
Response to Reviewer’s comment # 4: Thank you. Revised as requested.
Comment #5: Without basic information, it is not possible to draw a conclusion about genomic delineation of genes.
Response to Reviewer’s comment # 5: Thank you in advance for your consideration to our revised version of the manuscript.
Please consider that this is an original network-based study, an approach which is still uncommon to explore chromosomal imbalance rearrangements regions associated with ID. Systems biology approach is a new, promising approach to understand human diseases. The work is a documented evaluation of the efficiency of network-based analysis to study ID associated with chromosomal duplications. We have been demonstrating in recent publications(https://doi.org/10.1002/ajmg.c.31900;https://doi.org/10.3389/fgene.2020.00561; http://orcid.org/0000-0002-6397-1301), that the contribution of the network-based analysis is that, regardless of the origin of the pathogenesis of a deletion/duplication syndrome (epigenetic alteration, loss of function of the deleted proteins, effect of position, change of transcription factor sites or deregulation of miRNAs), it considers all these mechanisms when estimating how the proteins interact or fail to interact with each other in the interactome. Therefore, we propose that our manuscript may contribute to understand the impact of genomic imbalances sharing common significant biological processes associated with ID.
Further, we believe that this manuscript will be of interest to the readership of the journal because our study contribute to broaden our understading of contiguous gene imbalances and the approach reported here provides one more tool for the investigation of ID pathogenesis.
I look forward to hearing from you. In the meantime, please feel free to contact me if you need any additional information.
Round 2
Reviewer 3 Report
Comment #1: The authors should describe more clinical information about the three patients analyzed in this article
Response to Reviewer’s comment # 1: Suggestion accepted. A table was added in the manuscript.
The table 1 is satisfactory.
Comment #2: The authors did not show the 8q24.13q24.3, 18p11.32p11.21 and Xq22.3-q27.2 results in G-band, FISH, CMA, SNP or qPCR.
Response to Reviewer’s comment # 2: We agree with your comment. We confirmed that the genomic gains were investigated by cytogenetic conventional methods such G-band, FISH and CMA and, that they are duplications. Please consider that the cytogenetic characterization of the rearrangements are not the focus of the present study. We are exploring the content of coding genes involved in the duplications and a possible shared interaction between them (relationships among genes from these three different duplicated chromosomal regions). However, the summary of the CMA can be seen in Table 1 and S1.
I suggest an analysis of gene expression for this study, as there is no evidence about the association of gene expression and clinical finding. However, if the authors want to focus on the methodology o cytogenetic to the interactome and gene-disease association network, it’s essential to make very clear in the discussion and conclusion the need for further analyzes.
Comment #5: Without basic information, it is not possible to draw a conclusion about genomic delineation of genes.
Response to Reviewer’s comment # 5: Thank you in advance for your consideration to our revised version of the manuscript.
Please consider that this is an original network-based study, an approach which is still uncommon to explore chromosomal imbalance rearrangements regions associated with ID. Systems biology approach is a new, promising approach to understand human diseases. The work is a documented evaluation of the efficiency of network-based analysis to study ID associated with chromosomal duplications. We have been demonstrating in recent publications(https://doi.org/10.1002/ajmg.c.31900;https://doi.org/10.3389/fgene.2020.00561; http://orcid.org/0000-0002-6397-1301), that the contribution of the network-based analysis is that, regardless of the origin of the pathogenesis of a deletion/duplication syndrome (epigenetic alteration, loss of function of the deleted proteins, effect of position, change of transcription factor sites or deregulation of miRNAs), it considers all these mechanisms when estimating how the proteins interact or fail to interact with each other in the interactome. Therefore, we propose that our manuscript may contribute to understand the impact of genomic imbalances sharing common significant biological processes associated with ID. (This message should be included in the discussion).
The recent publication should be added to the references.
Author Response
Reply to the Review Report Round 2: Thank you for your comments and suggestions.
Comment #1: The authors should describe more clinical information about the three patients analyzed in this article.
Response to Reviewer’s comment # 1: Suggestion accepted. A table was added in the manuscript.
Reply from review to response to Reviewer’s comment # 1: The table 1 is satisfactory.
Comment #2: The authors did not show the 8q24.13q24.3, 18p11.32p11.21 and Xq22.3-q27.2 results in G-band, FISH, CMA, SNP or qPCR.
Response to Reviewer’s comment # 2: We agree with your comment. We confirmed that the genomic gains were investigated by cytogenetic conventional methods such G-band, FISH and CMA and, that they are duplications. Please consider that the cytogenetic characterization of the rearrangements are not the focus of the present study. We are exploring the content of coding genes involved in the duplications and a possible shared interaction between them (relationships among genes from these three different duplicated chromosomal regions). However, the summary of the CMA can be seen in Table 1 and S1.
Reply from review to response to Reviewer’s comment # 2: I suggest an analysis of gene expression for this study, as there is no evidence about the association of gene expression and clinical finding. However, if the authors want to focus on the methodology o cytogenetic to the interactome and gene-disease association network, it’s essential to make very clear in the discussion and conclusion the need for further analyzes.
Response to Reviewer’s reply comment # 2: I agree with your comment. Suggestion accepted. We make clear in the discussion and conclusion the need of further analysis.
Comment #5: Without basic information, it is not possible to draw a conclusion about genomic delineation of genes.
Response to Reviewer’s comment # 5: Thank you in advance for your consideration to our revised version of the manuscript. Please consider that this is an original network-based study, an approach which is still uncommon to explore chromosomal imbalance rearrangements regions associated with ID. Systems biology approach is a new, promising approach to understand human diseases. The work is a documented evaluation of the efficiency of network-based analysis to study ID associated with chromosomal duplications. We have been demonstrating in recent publications (https://doi.org/10.1002/ajmg.c.31900;https://doi.org/10.3389/fgene.2020.00561; http://orcid.org/0000-0002-6397-1301), that the contribution of the network-based analysis is that, regardless of the origin of the pathogenesis of a deletion/duplication syndrome (epigenetic alteration, loss of function of the deleted proteins, effect of position, change of transcription factor sites or deregulation of miRNAs), it considers all these mechanisms when estimating how the proteins interact or fail to interact with each other in the interactome. Therefore, we propose that our manuscript may contribute to understand the impact of genomic imbalances sharing common significant biological processes associated with ID.
Replies from review to response to Reviewer’s comment # 5:
- (This message should be included in the discussion).
Suggestion accepted. The paragraph was included in the discussion.
- The recent publication should be added to the references.
Response to Reviewer’s reply comment # 5: The recent publications were added to the references
Best regards,
Mariluce Riegel